# Realization of High Current Gain for Van der Waals MoS_2_/WSe_2_/MoS_2_ Bipolar Junction Transistor

**DOI:** 10.3390/nano14080718

**Published:** 2024-04-19

**Authors:** Zezhang Yan, Ningsheng Xu, Shaozhi Deng

**Affiliations:** State Key Laboratory of Optoelectronic Materials and Technologies, Guangdong Province Key Laboratory of Display Material and Technology, School of Electronics and Information Technology, Sun Yat-sen University, Guangzhou 510275, China; yanzzh3@mail2.sysu.edu.cn (Z.Y.); stsxns@mail.sysu.edu.cn (N.X.)

**Keywords:** two-dimensional material, van der Waals, bipolar junction transistor, vertical stacked, MoS_2_/WSe_2_/MoS_2_ heterostructure, breakdown characteristic

## Abstract

Two-dimensional (2D) materials have attracted great attention in the past few years and offer new opportunities for the development of high-performance and multifunctional bipolar junction transistors (BJTs). Here, a van der Waals BJT based on vertically stacked n^+^-MoS_2_/WSe_2_/MoS_2_ was demonstrated. The electrical performance of the device was investigated under common-base and common-emitter configurations, which show relatively large current gains of α ≈ 0.98 and β ≈ 225. In addition, the breakdown characteristics of the vertically stacked n^+^-MoS_2_/WSe_2_/MoS_2_ BJT were investigated. An open-emitter base-collector breakdown voltage (BV_CBO_) of 52.9 V and an open-base collector-emitter breakdown voltage (BV_CEO_) of 40.3 V were observed under a room-temperature condition. With the increase in the operating temperature, both BV_CBO_ and BV_CEO_ increased. This study demonstrates a promising way to obtain 2D-material-based BJT with high current gains and provides a deep insight into the breakdown characteristics of the device, which may promote the applications of van der Waals BJTs in the fields of integrated circuits.

## 1. Introduction

Since the discovery of graphene in 2004, a large number of two-dimensional (2D) atomic crystals, such as black phosphorus (BP), transition metal dichalcogenides (TMDs), and hexagonal boron nitride (h-BN), have been synthesized and extensively studied owing to their unique physical and chemical properties [1,2,3,4,5,6]. 2D materials have atomically thin thicknesses, which can be used to fabricate ultrathin and lightweight devices. Moreover, their dangling-bond-free surfaces make 2D layered materials very suitable to fabricate van der Waals heterojunctions as well as more complex layered architectures without considering the crystal lattice mismatch [7,8]. These fascinating properties make 2D materials, as well as their van der Waals heterostructures, have great application potential in high-performance photodetectors [9,10], memorizer [11,12], biosensors [13], gas sensors [14], and synaptic devices [15].

A bipolar junction transistor (BJT) is a two-junction, three-terminal semiconductor device that is widely used for signal amplification [16]. It is composed of three separately doped regions, namely, the emitter, base, and collector. The emergence of two-dimensional materials provides a platform for studying the electrical properties of BJTs with ultra-thin base regions, as the thickness of the base region has a great influence on device performance. In the past few years, several BJTs fabricated from 2D materials have been demonstrated with n-p-n [17] and p-n-p [18,19] configurations. Their electrical performance, as well as their potential applications such as photodetection [20,21] and biosensing [22], have been extensively studied. However, further improving the current gain and investigating the device breakdown behavior are still worthy of further research. A high-quality BJT usually consists of several components, such as asymmetric doping concentration, ultrathin base thickness, and high-quality contacts between semiconductors and metal electrodes. By systematically optimizing the structural design of the van der Waals BJT, there is still room to further improve the electrical performance of the device. 

In this study, we present a vertically stacked n^+^-MoS_2_/WSe_2_/MoS_2_ (n^+^-p-n) BJT. A re-doped MoS_2_ flake and an undoped MoS_2_ sheet were employed as the emitter and collector materials, respectively, to improve the current gain [22] and the breakdown voltage [23,24] of the van der Waals BJT. Also, we adopted appropriate n^+^-MoS_2_ and MoS_2_ thicknesses to minimize the impact of the Schottky barrier at the interface between 2D materials and metal electrodes. Relatively large current gains of α ≈ 0.98 and β ≈ 225 can be observed under common-base and common-emitter configurations. Additionally, the open-emitter base-collector breakdown voltage (BV_CBO_) and the open-base collector-emitter breakdown voltage (BV_CEO_) of the vertically stacked n^+^-MoS_2_/WSe_2_/MoS_2_ BJT under different temperatures were systematically explored, in which a positive temperature coefficient with the breakdown voltage can be observed.

## 2. Materials and Methods

### 2.1. Materials

The pristine MoS_2_ and WSe_2_ bulk crystals were purchased from HQ Graphene Company (Groningen, The Netherlands). The re-doped MoS_2_ bulk crystal was purchased from 2D Semiconductors (Scottsdale, AZ, USA).

### 2.2. Device Fabrication

The fabrication procedure of the vertically stacked n^+^-MoS_2_/WSe_2_/MoS_2_ BJT is illustrated in Figure 1. Firstly, a multilayer MoS_2_ flake was mechanically exfoliated onto the SiO_2_/Si (thickness of 300 nm and 525 μm, respectively) substrate by using scotch tape. Secondly, with the help of the micro-manipulator, a multilayer WSe_2_ flake stamped on polydimethylsiloxane (PDMS) film was transferred onto the previously prepared MoS_2_ flake to form a MoS_2_/WSe_2_ heterostructure at the overlapped region. Thirdly, through the same dry transfer process, a re-doped multilayer MoS_2_ flake was transferred onto the MoS_2_/WSe_2_ heterostructure to create a n^+^-MoS_2_/WSe_2_/MoS_2_ vertically stacked configuration. Then, the contact patterns were precisely defined using mask-less lithography, after which the Cr/Au electrodes (10 nm/100 nm) were deposited on the top of the TMD materials by thermal evaporation. Finally, annealing was performed at a temperature of 573 K for 2 h to remove the residual organic matter attached to the material interfaces.

### 2.3. Characterization

The composition and height profile of the vertically stacked n^+^-MoS_2_/WSe_2_/MoS_2_ BJT were characterized by Raman spectroscopy (In Via Reflex, Renishaw, Wotton-under-Edge, Gloucestershire, UK) with a 532 nm laser and atomic force microscopy (AFM) (NTEGRA Spectra, NT-MDT, Moscow, Russia), respectively. The temperature-dependent electrical characteristics of the device were carried out using a thermal system (ETC 200L, ESPEC, Osaka, Japan) and a semiconductor parameter analyzer (B1500A, Agilent Technologies, Santa Clara, CA, USA).

## 3. Results and Discussion

The vertically stacked n^+^-MoS_2_/WSe_2_/MoS_2_ BJT was fabricated by a controlled multistep dry transfer process. Here, the bottom MoS_2_ (top n^+^-MoS_2_) flake was determined as the collector (emitter) material, and the middle WSe_2_ flake was used as the base material. Figure 2a illustrates the Raman spectra of the individual materials and the heterojunction area. The blue line shows the Raman spectrum of the bottom MoS_2_ flake, in which the peaks at 384.3 cm^−1^ and 409.1 cm^−1^ can be observed [25]. For the WSe_2_ flake, it presents two Raman peaks centered at 249.8 cm^−1^ and 258.1 cm^−1^, consistent with previous works [26,27,28]. In addition, the Raman peaks in the MoS_2_/WSe_2_ overlapped region are consistent with the individual MoS_2_ and WSe_2_ sheets, indicating the successful preparation of the high-quality van der Waals heterostructure [29,30]. It should be noted that an appropriate increase in the thickness of the MoS_2_ and n^+^-MoS_2_ can improve the contact quality between 2D materials and metal electrodes, as shown in Appendix A. Therefore, relatively thick MoS_2_ (≈16.1 nm) and n^+^-MoS_2_ (≈17 nm) were employed as the collector and emitter regions, respectively. The thickness of WSe_2_, about 7.4 nm, was also characterized using AFM, as shown in Figure 2b.

To confirm the n-type and p-type behavior of the individual 2D materials, the transfer characteristics of the MoS_2_, WSe_2_, and re-doped MoS_2_ flakes were investigated, as shown in Appendix A. The pristine MoS_2_ and re-doped MoS_2_ show n-type behaviors with threshold voltages of around −4 V and −22.5 V, respectively, indicating the electron density of around 3.1 × 10^11^ cm^−2^ and 1.73 × 10^12^ cm^−2^, respectively. A p-type behavior can be observed in the WSe_2_ flake. Therefore, the formation of the n-p-n heterostructure is confirmed. The rectification properties of the base-emitter and base-collector junctions were also characterized, as shown in Appendix A. Negative differential resistance (NDR) effects were observed in the I–V curves of the base-emitter and base-collector junctions, which may be attributed to band-to-band tunneling [31].

The electrical performance of the vertically stacked n^+^-MoS_2_/WSe_2_/MoS_2_ BJT was first characterized under a common-base configuration. Here, the electrical measurements were performed in an ambient and dark environment. Figure 3a shows the schematic illustration of the electrical connection. The base-emitter junction was forward-biased, whereas the base-collector junction was reverse-biased. The base was connected to the ground. Figure 3b shows the input characteristics of the device, where the I–V curves of the emitter current (I_E_) versus the base-emitter voltage (V_BE_) at various fixed collector-emitter voltages (V_CB_) are performed. As the V_BE_ increases, more electrons can inject from the emitter to the base region owing to the lower barrier height at the base-emitter junction, leading to an increase in I_E_. Figure 3c shows the dependence of the collector current (I_C_) on the V_CB_ at various fixed V_BE_ values, namely the output characteristics of the device. At a small value of V_CB_ (V_CB_ < 1 V), only a portion of the electrons injected from the emitter into the base region can be successfully transmitted to the collector region. The increase in V_CB_ promotes the collector’s ability to drain electrons in the base region. Therefore, I_C_ increases approximately linearly with V_CB_. When V_CB_ is larger than 1 V, most of the electrons are injected into the collector region under the reverse bias of the base-collector junction, becoming the main component of the collector current. At this time, the value of I_C_ nearly becomes unaffected by changing V_CB_. The V_BE_ becomes the main factor affecting the collector current. Figure 3d shows the I_C_, I_E_, and the corresponding common-base current gain (α) as a function of V_BE_ at V_CB_ = 5 V. Both I_C_ and I_E_ increase exponentially with the increase in V_BE_. From the I–V curves of I_C_ and I_E_, the corresponding gain α (α = I_C_/I_E_) can be calculated to be approximately 0.98.

Then the performance of the vertically stacked n^+^-MoS_2_/WSe_2_/MoS_2_ BJT was investigated under a common-emitter configuration. Figure 4a shows the schematic illustration of the electrical connection. Figure 4b shows the input characteristics of the device, where the I–V curves of the base current (I_B_) versus V_BE_ at various fixed collector-emitter voltages (V_CE_) are performed. As the V_BE_ increases, the base current increases exponentially. Additionally, room-temperature NDR effects are observed in Figure 4b (red circle). As the V_BE_ increases, I_B_ initially increases but then decreases and finally gains a second increase in voltage, resulting in a negative slope on the current–voltage curve. This phenomenon may be attributed to lateral band-to-band tunneling, which is consistent with the previously reported performance of MoS_2_/WSe_2_ heterojunction [31]. The corresponding band diagram is shown in Appendix A. Figure 4c shows the dependence of V_CE_ on the collector current (I_C_) under several fixed V_BE_ values, which is called the output characteristic of the device. At a small value of V_CE_ (V_CE_ < 1.8 V), the collector current increases approximately linearly with V_CE_, which is defined as the saturation mode of the BJT. Above the saturation region, I_C_ is mainly dependent on V_BE_, indicating that the van der Waals BJT is operating in the active region. The difference between I_B_ and I_C_ with a fixed V_BE_ of 2 V is illustrated in Figure 4d. For a BJT operating under the common-emitter configuration, the current gain (β) can be calculated by taking the ratio of I_C_ and I_B_ [32]. A maximum common-emitter current gain of approximately 225 is obtained at V_CE_ = 6 V. The performance of the n-p-n BJTs fabricated by 2D materials is compared and listed in Table 1 [16,22,33,34,35]. Our van der Waals BJT exhibits a relatively high common-emitter current gain.

Furthermore, based on the prepared van der Waals BJT with excellent characteristics, the breakdown characteristics of the device were investigated. Figure 5a shows the open-emitter base-collector breakdown characteristics of the device at room temperature. In this case, the collector was set to ground, a reverse bias was applied to the base-collector junction, and the emitter electrode was set to an open state. At a reverse bias of <25 V, only a small reverse dark current of around 0.14 nA exists. As the reverse bias is increased beyond 25 V, a breakdown occurs, leading to a rapid increase in *I_CBO_*. The critical voltage (V_cr_) is determined to be 25 V (the gray line) when the dark current begins to increase rapidly [36]. The multiplication factor (*M*) can be calculated by taking the ratio of *I_CBO_* and *I_s_*, where *I_s_* represents the current at 25 V. Measurements of carrier multiplication *M* in junctions near breakdown lead to the following empirical equation [37]:(1)$$M=\frac{1}{1-{\left(V/V_{b}\right)}^{n}}$$
where *n* represents the ionization index and *V_b_* is a fitting parameter. Equation (1) can be rewritten as follows:(2)$$\ln(1-1/M)=n\ln(V/V_{b})$$
suggesting a linear dependence of ln(1−1/*M*) on ln(*V*) in the avalanche carrier multiplication effect. Figure 5b shows the relationship between ln(1−1/*M*) and ln(*V*), which directly indicates the occurrence of the avalanche breakdown. In addition, the avalanche breakdown voltage is sensitive to the temperature because a higher electric field is needed to compensate for the energy loss due to electron-phonon scattering as the temperature increases [38]. Therefore, a positive temperature coefficient of the breakdown voltage can be observed in the avalanche carrier multiplication effect [39,40]. Figure 5c shows the open-emitter base-collector breakdown characteristics of the device at different temperatures from 280 K to 340 K. The relationship between the BV_CBO_ (defined as the voltage at which an *I_CBO_* of 0.1 μA is measured) and temperature is consistent with the theoretical expectation (Figure 5d).

The open-base collector-emitter breakdown characteristics of the vertically stacked n^+^-MoS_2_/WSe_2_/MoS_2_ BJT under room temperature conditions were also investigated. The I–V curve and the schematic illustration of the electrical connection are shown in Figure 6a. The *I_CEO_* increases slowly at first and then rapidly increases when the V_CE_ is beyond V_cr2_ (=35 V), indicating that a breakdown occurs in the van der Waals BJT. The open-base collector-emitter breakdown characteristics of the van der Waals BJT at different temperatures from 280 K to 320 K were also investigated, as shown in Appendix A. The BV_CEO_ (defined as the voltage at which an I_CEO_ of 5 μA is measured) increases with the operating temperature, indicating a positive temperature coefficient of the breakdown voltage, as shown in Figure 6b.

## 4. Conclusions

In summary, a vertically stacked n^+^-MoS_2_/WSe_2_/MoS_2_ BJT was fabricated by a controlled multistep dry transfer process. The electrical characteristics of the device were investigated, which show relatively large current gains (α ≈ 0.98 and β ≈ 225) under common-base and common-emitter configurations. Additionally, to allow safe measurements at practical current densities, the temperature-dependent breakdown characteristics of the van der Waals BJT were investigated. A BV_CBO_ of 52.9 V and a BV_CEO_ of 40.3 V were obtained at room temperature. With the increase in the operating temperature, the breakdown voltage increases. This work provides a promising way to improve the electrical performance of the van der Waals BJT through proper architecture design and systematically demonstrates the temperature-dependence breakdown characteristics of the device, which might be helpful for the applications of the 2D-material-based BJT in the fields of integrated circuits.

## Figures and Tables

**Figure 1 nanomaterials-14-00718-f001:**
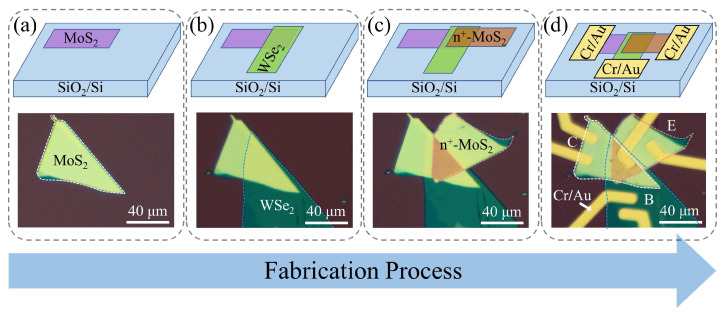
The schematic illustration and the optical microscopy images show the fabrication procedure of the vertically stacked n^+^-MoS_2_/WSe_2_/MoS_2_ BJT. (**a**) Bottom multilayer MoS_2_; (**b**) MoS_2_/WSe_2_ n-p heterostructure; (**c**) vertically stacked n^+^-MoS_2_/WSe_2_/MoS_2_ heterostructure; (**d**) Cr/Au metal electrodes connected to the heterostructure.

**Figure 2 nanomaterials-14-00718-f002:**
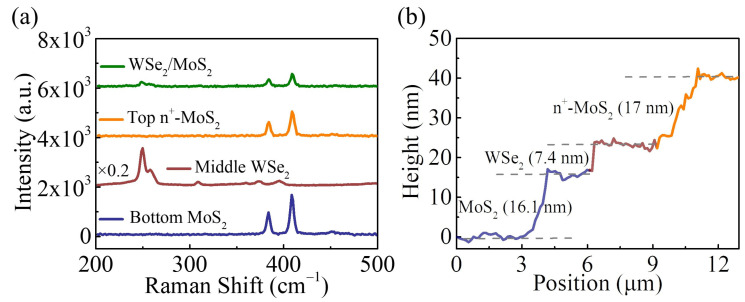
(**a**) Raman spectra of bottom MoS_2_ (blue line), WSe_2_ (brown line), top n^+^-MoS_2_ (orange line), and WSe_2_/MoS_2_ heterostructure (green line); (**b**) the AFM height profile of the BJT.

**Figure 3 nanomaterials-14-00718-f003:**
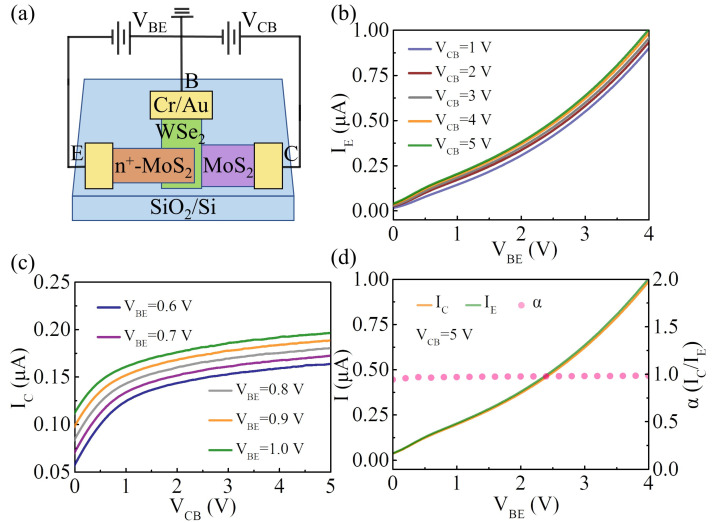
(**a**) Schematic illustration to show the electric connection of the device under the common-base configuration; (**b**) input characteristics of the device under the common-base configuration; (**c**) output characteristics of the device under the common-base configuration; (**d**) I_E_ (green curve), I_C_ (yellow curve), and α (pink dotted line) are shown as a function of V_BE_ at V_CB_ = 5 V.

**Figure 4 nanomaterials-14-00718-f004:**
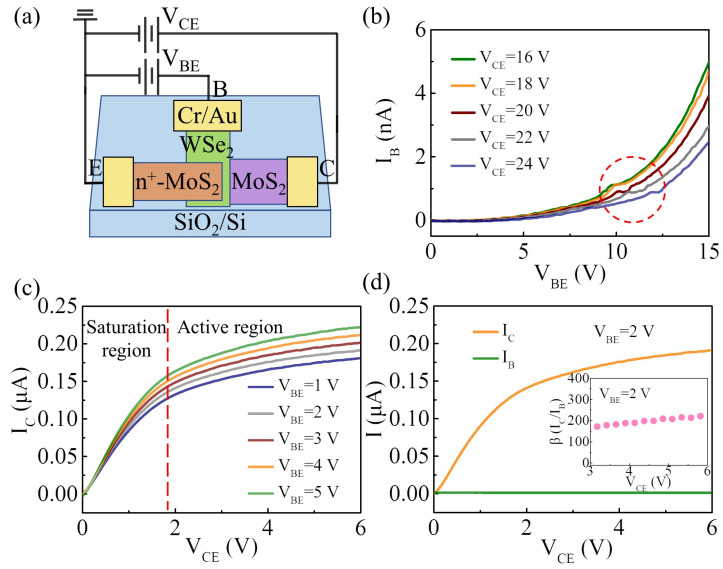
(**a**) A schematic illustration to show the electric connection of the device under the common-emitter configuration; (**b**) input characteristics of the device under the common-emitter configuration. The inset of the red circle shows the NDR effect region; (**c**) output characteristics of the device under the common-emitter configuration; (**d**) I_C_ (yellow curve) and I_B_ (green curve) are shown as a function of V_CE_ at V_BE_ = 2 V. The inset shows the calculated common-emitter current gain (β = I_C_/I_B_) as a function of V_CE_ at V_BE_ = 2 V.

**Figure 5 nanomaterials-14-00718-f005:**
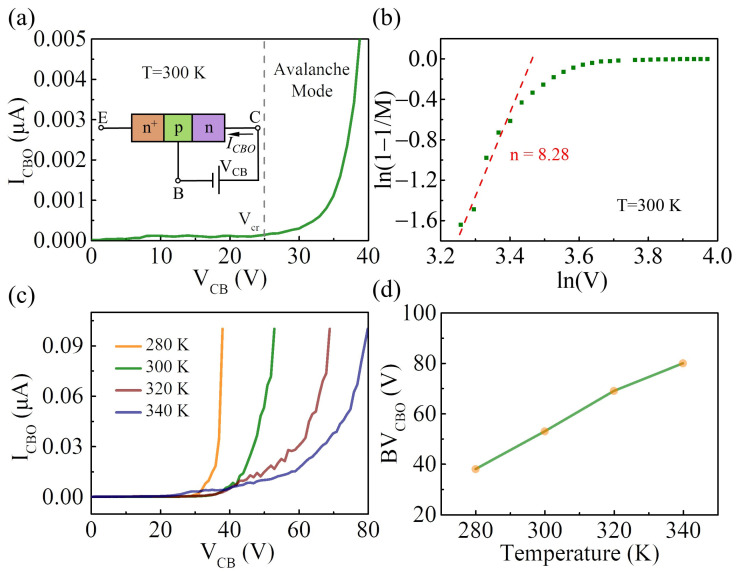
(**a**) I–V curve for the open-emitter base-collector junction showing the avalanche effect at a large reverse bias beyond V_cr_. The inset shows a schematic illustration of the electric connection; (**b**) logarithmic plot of 1−1/*M* versus *V*. *n* represents the ionization index, obtained by the linear fitting of the plot near ln(V_cr_); (**c**) I–V curve for the open-emitter base-collector junction at different temperatures (from 280 K to 340 K); (**d**) the BV_CBO_ as a function of temperature.

**Figure 6 nanomaterials-14-00718-f006:**
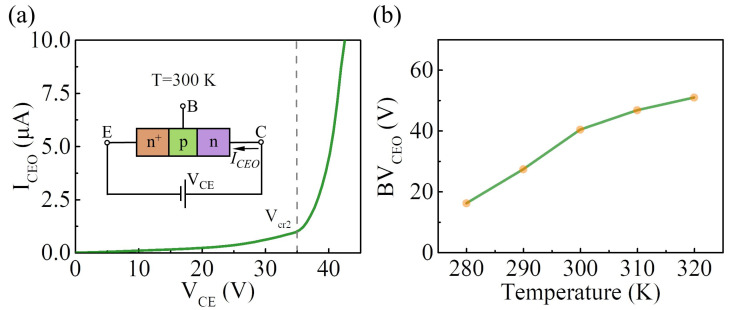
(**a**) The open-base collector-emitter breakdown characteristics of the van der Waals BJT at room temperature. The inset shows the schematic illustration of the electric connection; (**b**) the BV_CEO_ as a function of temperature.

**Table 1 nanomaterials-14-00718-t001:** Summary of the performance of this work and recently reported BJT prepared by 2D materials [16,22,33,34,35].

Material	Structure	Type	α	β	Ref.
n^+^-MoS_2_/WSe_2_/MoS_2_	Vertical	npn	0.98	225	This work
MoTe_2_/GeSe/MoTe_2_	Vertical	npn	0.95	29.3	[22]
MoS_2_/WSe_2_/MoS_2_	Vertical	npn	0.97	12	[34]
WS_2_/GeSe/WS_2_	Vertical	npn	1.11	20.7	[35]
MoS_2_/WSe_2_/MoS_2_	Lateral	npn	/	3	[33]
MoS_2_/BP/MoS_2_	Lateral	npn	0.98	41	[16]

## Data Availability

The data presented in this article are available at request from the corresponding author.

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
