# Peer review of "Realization of High Current Gain for Van der Waals MoS2/WSe2/MoS2 Bipolar Junction Transistor"

_nanomaterials, 2024, doi:10.3390/nano14080718_

Round 1
Reviewer 1 Report
Comments and Suggestions for Authors
The study details the creation and examination of a vertically stacked n+-MoS2/WSe2/MoS2 Bipolar Junction Transistor (BJT), demonstrating noteworthy current gains and temperature-dependent breakdown behaviors. Notably, under common-base and common-emitter configurations, current gains of approximately 0.98 and 225 are achieved, respectively. Moreover, the device reveals a breakdown collector-base open voltage (BV_CBO) of 52.9 V and a breakdown collector-emitter open voltage (BV_CEO) of 40.3 V at room temperature, with the breakdown voltage escalating at elevated temperatures. These insights underscore the potential of van der Waals BJTs to enhance the electrical efficacy of 2D-material-based integrated circuits, spotlighting the significance of architectural design and the impact of temperature on the stability and functionality of the device.
While the results are intriguing, the criticisms should be resolved to meet the criteria for publication in this journal. Specific criticisms are outlined below:
1. To ascertain the dominant carrier type in the WSe2 flake, its transfer characteristics should be measured. Despite the diode-like I-V characteristics presented for the n+-MoS2/WSe2 and MoS2/WSe2 junctions, such characteristics might also be observed in Schottky contacts between 2D materials and metal electrodes, as the non-linear I-V curves in Fig. S1 hint an existence of Schottky barriers at the MoS2-Cr/Au interfaces.
2. The flake's thickness influences the WSe2's characteristics; literature [Nano Res. 2018, 11, pp722–730] suggests varying behaviors based on thickness, with a 7.4 nm thickness possibly indicating n-type behavior in this study. Therefore, presenting the transfer characteristics of the WSe2 flake is crucial.
3. The manuscript references significant band-to-band tunneling but lacks experimental evidence for the claimed negative differential resistance (NDR), especially since the observed trends do not match those in the cited reference [21].
4. Furthermore, the appearance of small peaks around 2.5V in Fig. S3 raises questions about their origin, warranting an explanation.
5. While the high current gain in the common-emitter configuration is highlighted as a novelty, it's noted that higher gains have been reported elsewhere [ACS Photonics 2024, 11, pp649−659], suggesting the inclusion of current gain α in Table 1 for comparative purposes.
6. A minor correction is needed in line 27, where "hexanol boron nitride" should be amended to "hexagonal boron nitride." Lastly, the description of the Re doping process for MoS2 requires greater detail, particularly concerning doping concentration, to aid reader comprehension of the doping characteristics.
Comments on the Quality of English Language
The text is understandable in its current form, but there is room for enhancement.
Reviewer 2 Report
Comments and Suggestions for Authors
The paper presents an interesting experimental study on the fabrication of BJT based on 2D materials.
The study should be improved regarding the following comments:
(a) How is ensured that the overlap region between emitter and base region is on top of the collector, too? There must be some distance to avoid a shortage between emitter and collector. But emitter overlapping the base without a underlying collector will degrade current amplification.
(b) The NDR region in fig 4b is attributed to B2B tunneling. A sketch of the band diagram supporting this assumption at VBE =10V should support this conclusion.
Reviewer 3 Report
Comments and Suggestions for Authors
In this study the authors have present a van der Waals Bipolar Junction Transistor (BJT) utilizing vertically stacked n+-MoS2/WSe2/MoS2 layers. Electrical performance was evaluated under common-base and common-emitter configurations, demonstrating substantial current gains (α ≈ 0.98 and β ≈ 225). Investigation into breakdown characteristics revealed impressive open-emitter base-collector breakdown voltage (BVCBO) of 52.9 V and open-base collector-emitter breakdown voltage (BVCEO) of 40.3 V at room temperature, with further increases observed at higher operating temperatures. This research showcases a promising approach for developing 2D-materials-based BJTs with high current gains and offers valuable insights into device breakdown behavior, potentially advancing their application in integrated circuits.
Although this work seems very interesting, but some issues need to be addressed which are necessary for the improvement of this manuscript before accepting. My comments are as given below.
1. The introduction contains very general information. Authors should add detailed discussion about the gap and need of this study.
2. introduction section lacks the potential application and potential of 2D materials based van der waals heterostructure. Some recent studies should be highlighted as; Atomically engineered, high-speed non-volatile flash memory device exhibiting multibit data storage operations; A Novel Biosensing Approach: Improving SnS2 FET Sensitivity with a Tailored Supporter Molecule and Custom Substrate;Flexible Memory Device Composed of Metal-Oxide and Two-Dimensional Material (SnO2/WTe2) Exhibiting Stable Resistive Switching.
3. The more details about the recently reported bipolar junctions and their performance should be highlighted in the Table-1
4. The measurements details are not explained more clearly. Please add the details about the measuring instrument and proper connections to measure the Bipolar junction transistor.
5. More importantly, the stability of the device should be discussed? Is it measured in ambient environment or in vacuum?
6. English language need to be revised as there are some minor grammatical errors.
Comments on the Quality of English Language
NA
Round 2
Reviewer 1 Report
Comments and Suggestions for Authors
The authors have effectively addressed all of my comments, and the manuscript has been significantly improved by incorporating my review feedback. Therefore, I recommend publishing this manuscript in this journal.
Comments on the Quality of English LanguageThe English in this manuscript is sufficiently clear to understand the authors' presentations and arguments.